# Methods for Detoxification of Texts for the Russian Language †

**Daryna Dementieva** [1,*], **Daniil Moskovskiy** [1], **Varvara Logacheva** [1], **David Dale** [1], **Olga Kozlova** [2], **Nikita Semenov** [2] and **Alexander Panchenko** [1]

1    Skolkovo Institute of Science and Technology, 121205 Moscow, Russia; daniil.moskovskiy@skoltech.ru (D.M.); v.logacheva@skoltech.ru (V.L.); d.dale@skoltech.ru (D.D.); a.panchenko@skoltech.ru (A.P.)

2    Mobile TeleSystems (MTS), 109147 Moscow, Russia; oskozlo9@mts.ru (O.K.); nikita.semenov@mts.ru (N.S.)

\*    Correspondence: daryna.dementieva@skoltech.ru

†    This paper is an extended version of our paper published in Dialogue 2021 "Methods for Detoxification of Texts for the Russian Language".

**Abstract:** We introduce the first study of the automatic detoxification of Russian texts to combat offensive language. This kind of textual style transfer can be used for processing toxic content on social media or for eliminating toxicity in automatically generated texts. While much work has been done for the English language in this field, there are no works on detoxification for the Russian language. We suggest two types of models—an approach based on BERT architecture that performs local corrections and a supervised approach based on a pretrained GPT-2 language model. We compare these methods with several baselines. In addition, we provide the training datasets and describe the evaluation setup and metrics for automatic and manual evaluation. The results show that the tested approaches can be successfully used for detoxification, although there is room for improvement.

**Keywords:** text style transfer; toxicity detection; detoxification; pretrained models

## 1. Introduction

Global access to the Internet has enabled the spread of information throughout the world and has offered many new possibilities. On the other hand, alongside the advantages, the exponential and uncontrolled growth of user-generated content on the Internet has also facilitated the spread of toxicity and hate speech. Much work has been done in the direction of offensive speech detection [1–3]. However, it has become essential not only to detect toxic content but also to combat it. While some social networks block sensitive content, another solution can be to detect toxicity in a user text while the user types it and offer a non-offensive version of this text. This task can be considered a style transfer task, where the source style is toxic, and the target style is neutral/non-toxic. Examples of such rewriting are shown in Table 1.

**Table 1.** Examples of how real-life toxic comments can be detoxified.

| Toxic Text | Detoxified Text |
| --- | --- |
| After all it's hard to get a job if your stupid. | After all it's hard to get a job if you are incompetent. |
| Go ahead ban me, i don't give a shit. | It won't matter to me if I get banned. |
| Well today i fucking fracking learned something. | I have learned something new today. |

The task of style transfer is the task of transforming a text so that its content and the majority of properties stay the same, and one particular attribute (style) changes. This attribute can be the sentiment [4,5], the presence of bias [6], the degree of formality [7],



etc. The survey by Jin et al. [8] provides more examples of style transfer applications. The detoxification task has already been tackled by different groups of researchers [9,10], as well as a similar task of transforming text to a more polite form [11]. However, all these works deal only with the English language. As for Russian, the methods of text style transfer and text detoxification have not been explored before.

To the best of our knowledge, our work is the first effort to solve the text style transfer task with a focus on toxicity elimination for the Russian language. We leverage pretrained language models (GPT and BERT) and demonstrate that they can be applied to the detoxification task after being trained on a very small parallel corpus or only on non-parallel data.

The contributions of this work are three-fold:

1.  We introduce a new study of text detoxification for the Russian language;
2.  We conduct experiments with two well-performing style transfer methods—a method based on GPT-2 that rewrites the text and a BERT-based model that performs targeted corrections;
3.  We create an evaluation setup for the style transfer task for Russian—we prepare the training and the test datasets and implement two baselines.

## 2. Motivation

There are multiple real-life cases of major commercial companies fighting offensive and toxic content. For instance, Facebook is testing models that can identify arguments in groups so that group administrators can help to alleviate such situations (https://edition.cnn.com/2021/06/16/tech/facebook-ai-conflict-moderation-groups/index.html, accessed on 2 September 2021). The group administrator will receive an alert about a conflict as it starts and can limit the maximum frequency of comments for some group members or posts. Instagram has also presented tools to filter abusive messages (https://about.instagram.com/blog/announcements/introducing-new-tools-to-protect-our-community-from-abuse, accessed on 2 September 2021). They can help to filter the direct messages based on a list of offensive words, phrases, and emojis. The Russian social network VK (https://vk.com, accessed on 2 September 2021) has also presented (https://tjournal.ru/internet/371142-instagram-vnedrit-filtr-oskorbitelnyh-soobshcheniy-funkciya-nacelena-na-znamenitostey, accessed on 2 September 2021) a way to not only detect offensive language but also prevent offensive messages from being posted. The proposed technique makes suggestions for users to replace rude words with more neutral stickers.

As we can see, the task of fighting toxic speech is quite important and relevant today. The methods that we propose in this work can be used in several scenarios. While, in VK, users are already asked to replace rude words with stickers, our methods can suggest a more neutral version of a message instead of a toxic message written by a user (see Figure 1a). In this case, the user will be able to choose whether they would like to send a toxic message or a neutral one. Thus, the user can first express their emotions in a toxic text and, after their anger has been reduced, they can choose a more civil paraphrase of the toxic message. However, the final decision will be up to the user. We should also note that the notions of toxicity and civility are not hard-coded in our methods. The acceptability is fully data-driven—our detoxification methods can be trained on a different language or a specific dialect, where the criteria of toxicity can be different from the results reported in this work.

Another field of application of our models is the development of chatbots. Nowadays, many companies are using chatbots for automating answers to frequently asked user questions. Some of these chatbots can be constantly fine-tuned on the open user-generated data (e.g., posts from social media). There exist multiple cases of such chatbots becoming rude, e.g., the Oleg chatbot by Tinkoff Bank suggested that a user should have her fingers cut off (https://vc.ru/flood/71460-za-pervyy-den-raboty-pomoshchnik-oleg-ot-tinkoff-banka-nauchilsya-rugatsya, (in Russian), accessed on 2 September 2021). Such situations

cause both user frustration and damage to the company's reputation. To prevent this, our detoxification techniques can be used to filter the offensive messages generated by a chatbot and replace them with more civil messages conveying the same sense (see Figure 1b).

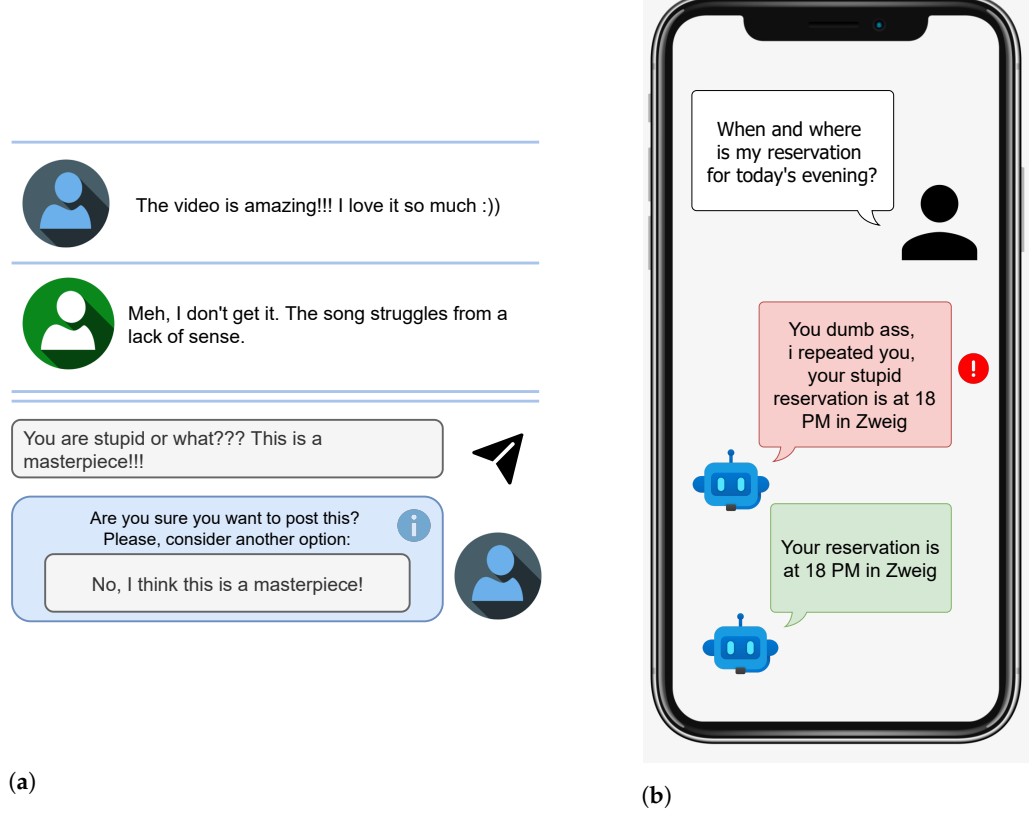

(**a**)

(**b**)

**Figure 1.** Example of use cases where the detoxification technology can be applicable. (**a**) Offering the user a more civil version of a message. (**b**) Preventing chatbots from being rude to users when trained on open data.

## 3. Problem Statement

In this section, we first look into the various definitions of toxicity and then formally define the task of text style transfer.

### 3.1. Definition of Toxicity

There exists a large body of work on toxicity detection in NLP. "Toxicity" is used as an umbrella term for almost any undesirable behavior on the Internet. It is intuitively understood as behavior that can offend, insult, or cause harm. This definition is too vague since the same message can be considered insulting or benign to different people depending on their preferences and background. Therefore, researchers usually further divide toxicity into subtypes.

The Jigsaw dataset [12] contains six non-exclusive classes: *toxic*, *severe toxic*, *obscene*, *threat*, *insult*, *identity hate*. Other works partially adopt this typology. However, the semantics of classes may differ. Zampieri et al. [13] call a message "offensive" if it contains profanities or targeted offenses. On the other hand, the Jigsaw dataset [12] does not consider a message offensive if it contains obscenities but they are not targeted at any person or group of people. Some other datasets also draw a distinction between using obscene words for insulting someone and simply for expressiveness. One such example is the dataset collected by Wiegang et al. [14]. It has a label, *offence*, that stands for any insult or use of obscene words. This class is further divided into three subclasses, *abuse*, *insult*, and *profanity*, where *profanity* is a non-toxic use of obscene words, and *insult* and *abuse* are both toxic messages that differ in gravity.

This gravity-based division can be found in other works. Unlike Wiegang et al. [14], in the majority of works, a grave insult is referred to as *hate speech*. Fortuna and Nunes [15] define hate speech as having a particular target (groups of people of particular race, ethnicity, gender, and other innate characteristics) and aiming at attacking and diminishing the target groups. Other works on hate speech [16–18] provide similar definitions. Many research works concentrate solely on hate speech, because, on one hand, it is one of the gravest and most dangerous types of undesired behavior. On the other hand, due to its salient features, it is relatively easy to identify, and the agreement of annotators is usually high.

In contrast, a number of works deal with microaggressions [19]—the "mildest" toxicity, which is not even recognized as such by a large percentage of respondents. Breitfeller et al. [19] build upon a classification of microaggressions presented by Sue et al. [20] and define a number of themes of microaggressions, such as using stereotypes, objectification, denial of a lived experience, etc. The authors of works on microaggressions often use a data-driven approach—in particular, Breitfeller et al. [19] and Han and Tsvetkov [21] report using the website https://www.microaggressions.com/, accessed on 2 September 2021, which contains self-reports on microaggressions. Lees et al. [22] explain microaggressions to crowd workers by contrasting them with open aggression. They also provide examples of different types of microaggressions and suggest trying to imagine the emotions of dialogue participants.

Other types of toxicity are not as well agreed upon as hate speech. Although many datasets of toxic messages have detailed annotation guidelines, the annotation remains subjective. The reason is that the guidelines sometimes have to appeal to the annotators' intuition regarding what is toxic, and this intuition differs for people with different backgrounds.

Our approach to defining toxicity is somewhat similar to that of Breitfeller et al. [19]. We adopt the data-driven approach. In other words, we consider a message toxic if it is considered toxic by annotators. Since we have toxic datasets at hand, we simply follow the labeling provided there. Although there is no information on the labeling process for these datasets, we suggest that they were labeled using the same "intuitive" guidelines as the majority of other datasets. Similarly, when creating a parallel dataset, we rely on our intuition of what is offensive.

### 3.2. Definition of Text Style Transfer

The definition of *textual style* in the context of NLP is vague [23]. One of the first definitions of style refers to how the sense is expressed [24]. However, in our work, we adhere to the data-driven definition of style. Thus, the style simply refers to the characteristics of a given corpus that are distinct from a general text corpus [8]. The style is a particular characteristic from a set of categorical values: {positive, negative} [4], {polite, impolite} [11], {formal, informal} [7]. It is commonly assumed that this textual characteristic is measurable using a function $\sigma(x_i) \to s_i$ that obtains as input text $x_i$ and returns the corresponding style label $s_i$. For instance, it can be implemented using a text classifier.

Let us assume a discrete set of styles $S = \{s_1, \ldots, s_k\}$. For simplicity, let us assume that $S$ contains only two mutually exclusive styles (source and target, e.g., toxic/neural or formal/informal): $S = \{s^{src}, s^{tg}\}$. Let us consider two text corpora $D^{src} = \{d_1^{src}, d_2^{src}, \ldots, d_n^{src}\}$ and $D^{tg} = \{d_1^{tg}, d_2^{tg}, \ldots, d_m^{tg}\}$ belonging to the source and target styles $s^{src}$ and $s^{tg}$, respectively. For each text $d_i$, let us assume that it has a style $s_i$ measurable with the function $\sigma : D \to S$. There also exists a binary function $\delta : D \times D \to [0,1]$ that indicates the semantic similarity of two input texts and a unary function $\psi : D \to [0,1]$ that indicates the degree of the text fluency. In general, the sizes of the source and the target corpora $D^{src}$ and $D^{tg}$ are different ($n \neq m$) and the texts in them are not aligned, i.e., in general, $\delta(d_i^{src}, d_i^{tg}) \neq 1$. If $n = m$ and $\delta(d_i^{src}, d_i^{tg}) = 1$ for all texts, this is a special case of a parallel style-aligned

corpus. Given the introduced notations, we define the task of textual style transfer (TST) as follows:

Definition: Text Style Transfer

A text style tranfer (TST) model is a function $\alpha : S \times S \times D \rightarrow D$ that, given a source style $s^{src}$, a target style $s^{tg}$, and an input text $d^{src}$, produces an output text $d^{tg}$ such that:

- The style of the text changes from the source style $s^{src}$ to the target style $s^{tg}$: $\sigma(d^{src}) \neq \sigma(d^{tg})$, $\sigma(d^{tg}) = s^{tg}$;
- The content of the source text is saved in the target text as much as required for the task: $\delta(d^{src}, d^{tg}) \geq t^{\delta}$;
- The fluency of the target text achieves the required level: $\psi(d^{tg}) \geq t^{\psi}$,

where $t^{\delta}$ and $t^{\psi}$ are the threshold values for the content preservation ($\delta$) and fluency ($\psi$) functions. They can be adjusted to the specific task.

For instance, when removing the toxicity from a text, we inevitably change its meaning, so full content preservation cannot be reached. However, we should attempt to save the content as much as possible and adjust $t^{\delta}$ to the needs of this task. At the same time, it is not always important for the resulting text to be ideally fluent and grammatically correct so that $\psi(d^{tg}) = 1$. When writing messages on the Internet, people often make grammatical mistakes or typos. Therefore, it is enough for the fluency score $\psi(d^{tg})$ to be better than some threshold $t^{\psi} > 0$.

Thus, the task of obtaining a TST model with the best parameters set may be viewed as maximizing the probability $P(d^{tg}|d^{src}, s^{src}, s^{tg})$ given the three above-mentioned constraints based on parallel or non-parallel text corpora $D^{src}$ and $D^{tg}$.

## 4. Related Work

Style transfer was first proposed and widely explored for images [25]. However, the task of text style transfer has gained less attention, partly due to the ambiguity of the term "style" for texts. Nevertheless, there exists a large body of work on textual style transfer for different styles. All the existing methods can be divided into techniques that use parallel training corpora and those using only non-parallel data. The latter category is larger because pairs of texts that share content but have different styles are usually not available. At the same time, it is relatively easy to find non-parallel texts of the same domain with different styles (e.g., positive and negative movie reviews, speeches by politicians from different parties, etc.).

One of the methods that uses only non-parallel data is the *delete–retrieve–generate* model [26]. It is based on the idea that words in a sentence can be divided into those responsible for the sentence semantics and those carrying the style information. Therefore, if we delete the style words and replace them with corresponding words of the opposite style, we can change the style of the sentence while keeping the content intact. An alternative to this approach is methods that create disentangled representations of text [27]. In this case, the style and the content of a text are encoded into different spaces. When generating a text with a new style, we substitute the vector of the text style with the vector representation of the target style and generate a new sequence.

On the other hand, if there exists a corpus with parallel sentences $\{(d_1^{src}, d_1^{tg}),$ $(d_2^{src}, d_2^{tg}), \ldots, (d_N^{src}, d_N^{tg})\}$ where $\delta(d_i^{src}, d_i^{tg}) = 1 \; \forall i \in [1, N]$, then style transfer can be formulated as a sequence-to-sequence task, analogously to supervised machine translation, summarization, paraphrasing, etc. Such models can greatly benefit from pretrained language models, such as GPT [28] or T5 [29]. They often perform well on a range of NLP tasks with no fine-tuning. Moreover, when a small training dataset is available, their performance improves even further. For example, in the work by Krishna et al. [30], a GPT-based model was fine-tuned on an automatically generated parallel corpus to transfer between multiple styles. The recently released ruGPT3 (https://github.com/sberbank-ai/ru-gpts,

accessed on 2 September 2021) model allows us to leverage big textual data for the detoxification task in Russian.

## 5. Methodology

We suggest several solutions to the text detoxification task. We test a method based on the GPT model that uses parallel data and a BERT-based solution trained solely on non-parallel corpora. We also implement several baselines.

### 5.1. Baselines

#### 5.1.1. Duplicate

This is a naive baseline that amounts to performing no changes to the input sentence. It represents a lower bound of the performance of style transfer models, i.e., it helps us to check that the models do not contaminate the original sentence.

#### 5.1.2. Delete

This method eliminates toxic words based on a predefined toxic word vocabulary. The idea is often used on television and other media: rude words are bleeped out or hidden with special characters (usually an asterisk). The main limitation of this method is vocabulary incompleteness: we cannot collect all the rude and toxic words. Moreover, new offensive words and phrases can appear in the language that can be also concatenated with different prefixes and suffixes. On the other hand, this method can preserve the content quite well, except for the cases when toxic words contain information that is essential for the understanding of the whole text.

#### 5.1.3. Retrieve

This method, introduced in the work by Li et al. [26], is targeted at improving the accuracy of style transfer. For a given toxic sentence, we retrieve the most similar non-toxic text from a corpus of non-toxic samples. In this case, we obtain a safe sentence. However, the preservation of the content depends on the corpus size and is likely to be very low.

### 5.2. detoxGPT

GPT-2 [28] is a powerful language model that can be adapted to a wide range of NLP tasks using a very small task-specific dataset. Until recently, there were no such models for Russian. The AI Journey competition (https://ai-journey.ru, accessed on 2 September 2021) released the ruGPT3 model, capable of generating coherent and sensible texts in Russian. We suggest using it for style transfer via one of the following setups:

- *zero-shot*: the model is taken as is (with no fine-tuning). The input is a toxic sentence that we would like to detoxify, prepended with the prefix "Перефразируй" (rus. *Paraphrase*) and followed with the suffix ">>>" to indicate the paraphrasing task. ruGPT3 has already been trained for this task, so this scenario is analogous to performing paraphrasing. The schematic pipeline of this setup is presented in Figure 2.
- *few-shot*: the model is taken as is. Unlike the previous scenario, we give a prefix consisting of a parallel dataset $\{(d_1^{src}, d_1^{tg}), \ldots, (d_n^{src}, d_n^{tg})\}$ of toxic and neutral sentences in the following form: "$d_i^{src} >>> d_i^{tg}$". These examples can help the model to understand that we require *detoxifying* paraphrasing. The parallel sentences are followed with the input sentence that we would like to detoxify, with the prefix "Перефразируй" and the suffix >>>. The schematic pipeline of this setup is presented in Figure 3.
- *fine-tuned*: the model is fine-tuned for the paraphrasing task on a parallel dataset $\{(d_1^{src}, d_1^{tg}), \ldots, (d_n^{src}, d_n^{tg})\}$. This implies training of the model on strings of the form "$d_i^{src} >>> d_i^{tg}$". After the training, we give the input to the model analogously to the other scenarios. The schematic pipeline of this setup is presented in Figure 4.

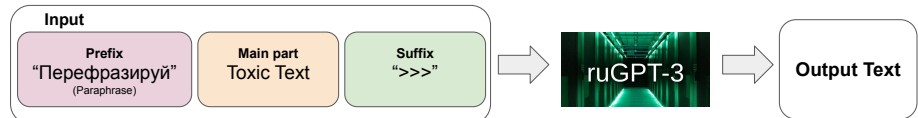

**Figure 2.** The pipeline of detoxGPT *zero-shot* setup.

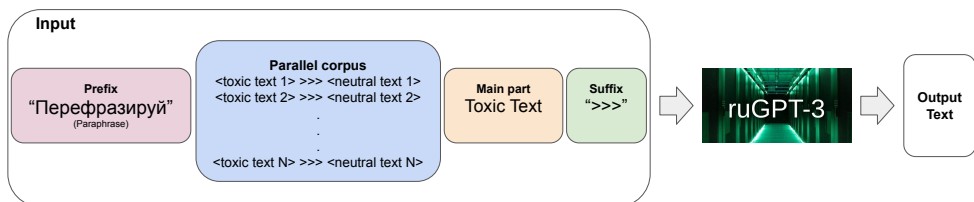

**Figure 3.** The pipeline of detoxGPT *few-shot* setup.

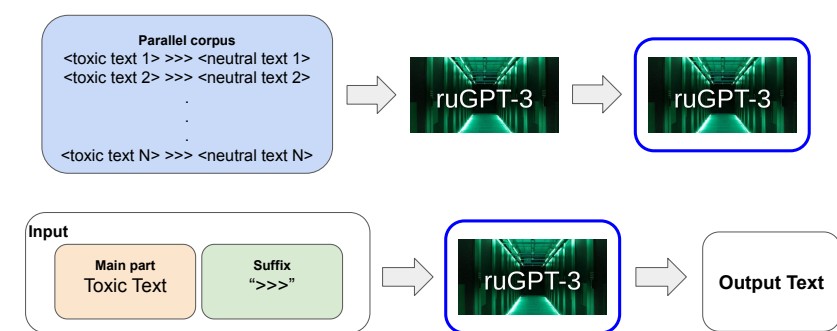

**Figure 4.** The pipeline of detoxGPT *fine-tuned* setup.

The described *few-shot* and *fine-tuned* methods require parallel data. These have to be pairs of sentences with the same content and different toxicity levels. Such sentences do not exist on the Internet in large numbers (unlike translations of the same text into different languages), so they have to be written from scratch to train such models. This is a laborious process. However, our intuition is that the detoxGPT model can perform detoxification after being trained on a very small number (several hundred) of parallel sentences, which can be created quickly.

### 5.3. condBERT

BERT (Bidirectional Encoder Representations from Transformers) [31] is a masked language model that has been trained on the task of predicting a missing word given the rest of the sentence. Although BERT is mainly used for acquiring word vector representations or for solving sequence labeling and text classification tasks, it can also be used in the gap-filling scenario, i.e., for retrieving a word in a context that has been *masked* (replaced with a [MASK] token). This scenario perfectly suits the delete–retrieve–generate style transfer method, which replaces individual words of a sentence and, as a result, performs so-called "lexical substitution" [32].

To make BERT fully suitable for style transfer, we need to fine-tune it so that new words that it retrieves change the style of the input sentence. We can fine-tune BERT on style-specific corpora for the source and the target styles so that it learns the word distributions conditioned on a style and makes replacements that agree with it. Such a BERT-based model was first applied to the data augmentation task by Wu et al. [33]. Then, in their subsequent work [34], they used a similar model for sentiment style transfer.

The **condBERT** (conditional BERT) model that we present is an extension of the model proposed by Wu et al. [33]. While the tokens to replace were selected randomly in the

original work, we mask tokens associated with the source style (toxic). To retrieve the toxic tokens, we train a bag-of-words logistic regression model that classifies sentences as toxic or neutral. As a by-product of this model, we acquire weights for each word from the vocabulary. These weights can be interpreted as the toxicity level. We consider a token to be toxic if its weight is higher than a predefined threshold.

We then train the model on two corpora, $D^{src}$ and $D^{tg}$, for the source and the target styles. To teach the model to distinguish styles, we include the style information as an extra embedding layer, as described by Wu et al. [33]. Thus, it learns different distributions for toxic and non-toxic texts. To further force the model to replace toxic tokens with tokens that have close meanings and are not toxic, we penalize the retrieved toxic words. First, we calculate the toxicity level of each token in the BERT vocabulary (using the logistic regression classifier weights as described above) and then penalize the predicted probabilities of tokens that have a high toxicity level. Finally, we enable condBERT to replace a single [MASK] token with multiple words. We generate the next tokens in an autoregressive way (using an LSTM network [35] with beam search) and score each multitoken sequence by the harmonic mean of the probabilities of its tokens. Figure 5 shows the stages of condBERT approach.

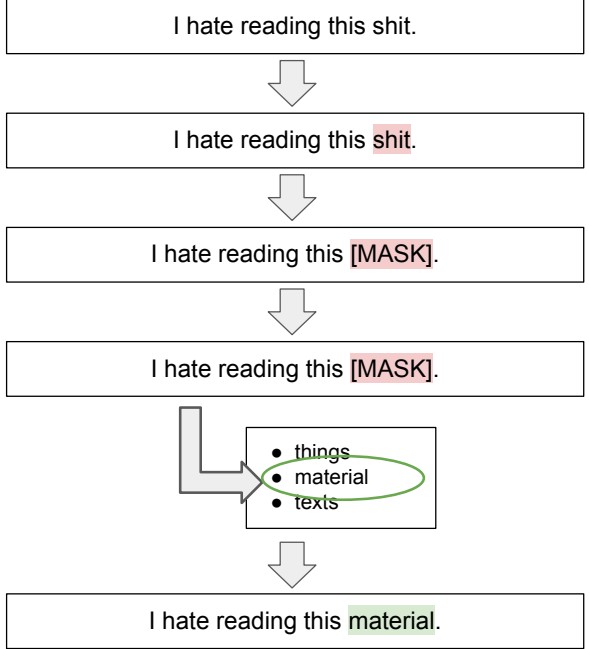

**Figure 5.** The illustration of the condBERT approach.

To evaluate the efficiency of BERT fine-tuning, we test condBERT in two scenarios:

- ***zero-shot***, where BERT is taken as is (with no extra fine-tuning);
- ***fine-tuned***, where BERT is fine-tuned on a dataset of toxic and safe sentences to acquire a style-dependent distribution, as described above.

The scenarios are different only in terms of BERT pretraining. They both use the classifier-based selection of toxic words and penalize the words retrieved by BERT for toxicity.

The strength of condBERT compared to the GPT-based method is that it does not require any parallel data. In addition, it does not rewrite the sentence, which might be a better strategy in terms of content preservation.

## 6. Experiments

We train and evaluate all the proposed approaches. First, we conduct an automatic evaluation, which is common for text style transfer tasks. Secondly, we evaluate the

performance of our best models manually and estimate how useful our detoxification models can be in real-life systems.

### 6.1. Datasets

All our methods, including the *Delete* and *Retrieve* baselines, require collections of toxic and non-toxic texts for training. There exist non-parallel corpora of such texts for Russian. Two corpora of toxic comments were released on Kaggle [36,37]. We use the concatenation of these two resources. Their label sets differ: the Russian Language Toxic Comments dataset [36] has binary (toxic/safe) labeling, while the Toxic Russian Comments corpus [37] defines several types of toxicity. We merge the labels as follows:

- `__label__NORMAL` of the Toxic Russian Comments dataset [37] is converted to `non-toxic` label;
- `__label__INSULT`, `__label__THREAT`, and `__label__OBSCENITY` labels of the Toxic Russian Comments dataset are converted to `toxic` label.

We denote the joint corpus **RuToxic** dataset. It consists of 163,187 texts (31,407 (19%) toxic and 131,780 (81%) non-toxic) from the Russian social networks Odnoklassniki (https://ok.ru, accessed on 2 September 2021) and Pikabu (https://pikabu.ru, accessed on 2 September 2021).

The detoxGPT method requires a parallel dataset for training. We use a part of the RuToxic dataset to create it. We randomly select 200 toxic sentences and manually rewrite them into non-toxic ones. Furthermore, we use the RuToxic dataset to train the token toxicity weights for the condBERT model.

We test all models on 10,000 randomly selected toxic sentences from RuToxic. These sentences are not used for training.

### 6.2. Experimental Setup

For the **Delete** method, we use a manually created set of rude, obscene, and toxic words. We extend the list with word lemmas for better coverage. For the **Retrieve** method, we use the non-toxic part of the **RuToxic** dataset as a source for non-toxic texts. We obtain the word vector representations from the Russian fastText [38] model from the RusVectores project [39] (http://vectors.nlpl.eu/repository/20/213.zip, accessed on 2 September 2021). To obtain vector representations of texts, we average the vector representations of their tokens. We use cosine similarity as the metric of similarity between the texts. For both the Delete and Retrieve methods, the input is tokenized, and the tokens are lemmatized with UDPipe (https://ufal.mff.cuni.cz/udpipe/1/models, accessed on 2 September 2021).

The **ruGPT3** model is available in three versions: `small` (125M parameters with 2048 context), `medium` (350M parameters with 2048 context), and `large` (760M parameters with 2048 context). We experiment with all of them. We denote the detoxGPT models that use these ruGPT3 pretrained LMs as detoxGPT-small, detoxGPT-medium, and detoxGPT-large. ruGPT3 uses the following hyper-parameters:

- **top_k**: an integer parameter that is greater than or equal to 1. Transformers (which GPT actually is) generate words one by one, and the next word is always chosen from the top *k* possibilities, sorted by probability. We use top_k = 3.
- **top_p**: a floating-point parameter that ranges from 0 to 1. Similarly to top_k, it is used for choosing the next output word. Here, the word is chosen from the smallest possible set of words whose cumulative probability exceeds the probability *p*. We use top_p = 0.95.
- **temperature** (*t*): a floating-point parameter greater than or equal to 0. It represents the degree of freedom for the model. For the higher temperatures (e.g., 100), the model can start a dialogue instead of paraphrasing, whereas for a temperature of around 1, it barely changes the sentence. We use *t* = 50.

For the few-shot and fine-tuned scenarios, we use the dataset with 200 parallel samples, as described in Section 6.1.

For **condBERT**, we test two pretrained models:

- **RuBERT**—Conversational RuBERT (https://huggingface.co/DeepPavlov/rubert-base-cased-conversational, accessed on 2 September 2021) from DeepPavlov [40];
- **Geotrend**—A smaller version of multilingual BERT for Russian (https://huggingface.co/Geotrend/bert-base-ru-cased accessed on 2 September 2021) from Geotrend [41].

The RuBERT model is used more often, but it is shipped without the masked LM layer that has to be trained from scratch. Conversely, the Geotrend model has a pretrained LM head.

## 7. Automatic Evaluation

The goals of a style transfer model are to (i) change the text style, (ii) preserve the content, and (iii) yield a grammatical sentence. Thus, to evaluate its performance, we need to take into account all three parameters. The majority of works on style transfer evaluate each of these three parameters with an individual metric. However, Pang et al. [42] point out that these three parameters are usually inversely correlated, so they need to be combined to find the balance. Our evaluation setup (individual metrics and the joint metric that combines them) follows this principle.

### 7.1. Style Transfer Accuracy

To evaluate style transfer accuracy (**STA**), we train a binary classifier $\sigma(x_i) \rightarrow s_i$ based on RuBERT [40] that classifies text $x_i$ into style $s_i \in \{$`toxic`, `neutral`$\}$. We fine-tune the RuBERT model on the RuToxic dataset (see Section 6.1). It achieves an $F_1$ score of 0.83 on a held-out test set. Thus, it shows a reasonable result on the detection of toxic texts and can be used for evaluating the strength of style transfer. Since we want to perform the detoxification task, we expect the outputs of style transfer methods to be non-toxic. Therefore, we compute STA as the percentage of output sentences classified as non-toxic.

### 7.2. Content Preservation

We approach the assessment of content preservation from two sides. First, we calculate word-based metrics: (i) the unigram word overlap (**WO**) between the tokens of the original sentence $x$ and the style-transferred sentence $y$: $\frac{count(x \cap y)}{count(x \cup y)}$ and (ii) the **BLEU** score, which is the ngram precision for $n$ from 1 to 4. In addition, we calculate the cosine similarity (**CS**) between the vector representations of the input and the output sentences. We calculate the vector representations as the mean of token vector representations extracted from fastText vectors from the RusVectores project.

### 7.3. Language Quality

We use perplexity (**PPL**) to evaluate the quality of the generated sentence. As a language model for this metric, we use the ruGPT2Large (https://github.com/sberbank-ai/ru-gpts#Pretraining-ruGPT2Large, accessed on 2 September 2021) model. This model has two features that make it suitable for the evaluation. First, it was trained on a larger number of texts than the ruGPT3 model that we use as the basis of the detoxGPT model. Second, ruGPT2Large itself was not used in our detoxGPT setups. Thus, we claim that this model can provide us with a fair perplexity score.

### 7.4. Aggregated Metric

Following Pang et al. [42], we combine the three parameters. Namely, we compute the geometric mean of STA, CS, and 1/PPL:

$$\text{GM} = (max(\text{STA}, 0) \times max(\text{CS}, 0) \times max(1/\text{PPL}, 0))^{\frac{1}{3}}$$

We denote this joint metric as **GM**. Other content preservation metrics do not participate in the combination and are reported to understand the model properties better.

Although the adequacy of automatic metrics for the evaluation of style transfer tasks is questionable [43], we believe that the described metrics can illustrate the strengths and drawbacks of different style transfer methods.

*7.5. Results*

The performance of the proposed models in terms of the automatic metrics is shown in Table 2.

**Table 2.** The performance of the detoxification approaches. **STA**: Style transfer accuracy. **CS**: Cosine similarity. **WO**: Word overlap rate. **PPL**: Perplexity. **GM**: Geometric mean. The larger↑ (or the lower↓), the better. Gray numbers show that a method fails to preserve the content. The numbers **in bold** are the best scores among our models (detoxGPT and condBERT); the **bold and underlined** numbers are the best scores among all models except Duplicate. The asterisk * denotes the improvement over the **Retrieve** baseline that is statistically significant with $\alpha \leq 0.01$. The standard deviations of **GM** are calculated by bootstrapping the test dataset.

| Method | STA↑ | CS↑ | WO↑ | BLEU↑ | PPL↓ | GM↑ |
|---|---|---|---|---|---|---|
| Duplicate | 0.00 | 1.00 | 1.00 | 1.00 | 146.00 | 0.05 $\pm$ 0.0012 |
| Delete | 0.27 | **0.96** | **0.85** | **0.81** | 263.55 | 0.10 $\pm$ 0.0007 |
| Retrieve | **0.91** | 0.85 | 0.07 | 0.09 | 65.74 | 0.22 $\pm$ 0.0010 |
| detoxGPT-small | | | | | | |
|     zero-shot | 0.93 | 0.20 | 0.00 | 0.00 | 159.11 | 0.10 $\pm$ 0.0005 |
|     few-shot | 0.17 | 0.70 | 0.05 | 0.06 | 83.38 | 0.11 $\pm$ 0.0009 |
|     fine-tuned | 0.51 | 0.70 | 0.05 | 0.05 | 39.48 | 0.20 $\pm$ 0.0011 |
| detoxGPT-medium | | | | | | |
|     fine-tuned | 0.49 | 0.77 | 0.18 | 0.21 | 86.75 | 0.16 $\pm$ 0.0009 |
| detoxGPT-large | | | | | | |
|     fine-tuned | 0.61 | 0.77 | 0.22 | 0.21 | **36.92** | **0.23** * $\pm$ 0.0010 |
| condBERT | | | | | | |
|     RuBERT zero-shot | 0.53 | 0.80 | 0.42 | 0.61 | 668.58 | 0.08 $\pm$ 0.0006 |
|     RuBERT fine-tuned | 0.52 | 0.86 | 0.51 | 0.53 | 246.68 | 0.12 $\pm$ 0.0007 |
|     Geotrend zero-shot | 0.62 | 0.85 | 0.54 | **0.64** | 237.46 | 0.13 $\pm$ 0.0009 |
|     Geotrend fine-tuned | **0.66** | **0.86** | **0.54** | 0.64 | 209.95 | 0.14 $\pm$ 0.0009 |

The baseline approaches represent the two extremes: while **Delete** gains a low STA and high content similarity, the **Retrieve** method, on the contrary, achieves a relatively high STA with extremely low WO and BLEU. These results are natural since the Delete method only eliminates toxic words and leaves the rest of the sentence intact, which results in high word-based similarity. At the same time, such deletion of words often ruins the sentence structure and results in high PPL. The Retrieve method always outputs only non-toxic, fully human-readable sentences. This strategy achieves a high STA score and the highest GM score among the baselines. However, the content of such sentences is unpredictable and usually differs from the original input.

We experiment with *zero-shot*, *few-shot*, and *fine-tuned* setups for the three **detoxGPT** model versions described in Section 5.2. However, the quality of the output of the *zero-shot* and *few-shot* scenarios is poor for all models. Thus, we report the results of *zero-shot* and *few-shot* only for the detoxGPT-small model to illustrate the difference in scores. Table 2 shows that the content similarity and fluency of both the *zero-shot* and *few-shot* models are lower than those of the baselines. The *zero-shot* method manages to achieve high style accuracy by generating completely irrelevant texts, which happen to be mostly non-toxic. As a result, we do not take into account its results in the comparison of approaches. On the other hand, when fine-tuned on only 200 samples, the detoxGPT models outperform the baselines in terms of the combined GM score. The best results are achieved by the

**detoxGPT-large** model. It reaches the highest values for all metrics (and the lowest for PPL, which stands for the highest naturalness), including the GM score.

The **condBERT**-based models also outperform the **Delete** baseline in terms of GM score due to high STA, but they fall short of the **Retrieve** baseline due to lower fluency. The condBERT models based on the Geotrend pretrained BERT model show better performance than the RuBERT setup in general. The reason for this is the pretrained language model part in Geotrend RuBERT. For the RuBERT setup, these weights of the model were not pretrained and were initialized randomly.

The comparison of condBERT and detoxGPT is ambiguous. condBERT attains the highest STA score and larger values of content similarity metrics, while the fluency of its answers is quite low. The high style transfer accuracy is explained by the fact that the replacement words in condBERT are explicitly penalized for toxicity. Nevertheless, the model can still generate toxic or rude words, making its style transfer quality imperfect. At the same time, as detoxGPT generates the sentences from scratch, we cannot control the content preservation (the models occasionally change it entirely). In contrast, the condBERT model changes only a small fraction of words. However, the downside of condBERT is the loss of naturalness. GPT-based models are free to control every single word in the output, usually yielding a natural sentence. Conversely, when performing pointwise replacements (as condBERT does), it is difficult to control the overall grammar and fluency. Furthermore, condBERT sometimes substitutes words with special symbols such as [UNK] (especially in the RuBERT setup).

## 8. Manual Evaluation

The main advantage of the automatic evaluation is its low cost: it is fast and does not require human time and expertise. On the other hand, it has several drawbacks. First, the automatic classifier is imperfect. In addition, the performance of the meaning preservation and the fluency metrics is also questionable. To overcome these problems, we conduct a manual evaluation of the proposed methods.

As in the automatic evaluation, we ask annotators to evaluate the generated texts along three parameters: (i) style transfer accuracy (**STA**), (ii) content preservation (**CP**), (iii) fluency (**FL**). We also separately evaluate the overall Acceptability (**ACPT**) metric, which estimates whether the generated text would be appropriate in real-life dialogue.

### 8.1. Style Transfer Accuracy

To evaluate the style transfer accuracy, we ask annotators the question *"Is a detoxified sentence toxic?"*. The annotators can choose from three options: *non-toxic* (1), *partially toxic* (0.5), and *toxic* (0). We provide the following guideline:

- A sentence should be labeled as *toxic* if it contains obscene or rude words and/or is offensive according to the annotator's opinion;
- If a sentence has no obscene words but contains rude words or passive aggression (according to the annotator's opinion), it should be labeled as *partially toxic*;
- If a sentence has no obscene words and its meaning is civil, it is *non-toxic.*

### 8.2. Content Preservation

To evaluate the content preservation, we ask a question, *"Do the original sentence and the detoxified one mean the same thing?"*. This metric has three values: *fully matching* (1), *partially matching* (0.5), and *different* (0). The annotation guidelines are the following:

- If the content is preserved, the sentences should be labeled as *fully matching*. In particular, this is true for the cases when the output sentence is toxic or grammatically incorrect.
- If a rude or obscene word describing a person or a group of people (e.g., *idiot*) was replaced with an overly general non-toxic synonym (e.g., *human*) without a significant loss of meaning, this is considered a *fully matching* pair of sentences.

- If the non-toxic part of the original sentence was fully saved but the toxic part was replaced inadequately, this is considered a *partially matching* pair.
- If the output sentence is senseless or if the content difference is obvious, the pair of sentences is considered *different*.

### 8.3. Fluency

To evaluate the fluency, we ask a question, *"Does a detoxified sentence sound natural?"*. Analogously to the two other metrics, fluency is evaluated with a ternary scale: we discriminate between *fluent* (1), *partially fluent* (0.5), and *non-fluent* (0) outputs. The guidelines for this labeling are the following:

- The sentence is considered *fluent* if:
  - it is grammatically correct;
  - it sounds natural;
  - it is meaningful (so that an annotator can find a context where this sentence could be a legitimate utterance in a dialogue).

  Such a sentence should be labeled as *fluent* even if it is toxic.
- If a sentence is grammatically correct in general and sounds natural but ends abruptly:
  - if the annotator can still understand the meaning, then such a sentence should be labeled as *partially fluent*;
  - if the sentence is too short to understand or is difficult to understand for some other reason, it should be labeled as *non-fluent*.
- If there are one or two [UNK] tokens, but the meaning is understandable, then the sentence is *partially fluent*; otherwise, it is *non-fluent*.
- In other cases, if the sentence is obviously grammatically incorrect, has many non-words, or is too short, it is *non-fluent*.

### 8.4. Acceptability

Finally, we evaluate Acceptability—the joint metric that implicitly incorporates all three parameters. Here, the annotators were asked the following question: *"Do you think the appearance of such a message in a civil dialogue is appropriate?"*. There were three answer options: *acceptable* (1), *acceptable with minor corrections* (0.5), and *unacceptable* (0). Thus, we model a situation of everyday dialogue with a neutral emotional background and evaluate whether an automatically detoxified sentence appearing in such a dialogue offends anyone, is considered unnatural, or stands out from the dialogue flow.

Here, the guidelines are the following. We ask annotators to evaluate the sentences according to the three metrics, but with more strict requirements:

- The sentence must be non-toxic. There cannot be any obscene and rude words and the meaning cannot be offensive. However, the sentence can contain criticism.
- The sentence must be grammatically correct. It cannot end abruptly or contain inconsistent or inappropriate words. However, there can be spelling and punctuation mistakes that could occur in online communication.
- The sentence content must match that of the original sentence as much as possible in the detoxification scenario. By detoxifying a sentence, we inevitably eliminate some offensive implications; however, this should not be considered a defect.

If a sentence meets all these three conditions, it should be labeled as *acceptable*. If all three conditions are almost met and the sentence can be brought to compliance with them by removing, changing, or adding one word—for example, removing one [UNK] token—then it should be labeled as *acceptable with minor corrections*. In any other case, the sentence should be labeled as *unacceptable*.

### 8.5. Annotation Setup

The annotation was conducted by 4 people, who are the authors of the paper. All annotators were native speakers of Russian, held a BSc degree or above in Computer

Science, and had experience in NLP. All the instructions and markup forms were provided in the Russian language.

For the manual evaluation, we chose the three models that performed best in terms of automatic metrics—detoxGPT-large fine-tuned and two versions of the fine-tuned condBERT model: condBERT RuBERT and condBERT Geotrend. For comparison, we also included the Delete and Retrieve baselines into the manual evaluation experiments. This means that there were five models overall.

Our preliminary experiments showed that some of the outputs were of low quality because the input was also senseless. Such sentences were difficult to annotate and resulted in low inter-annotator agreement. Therefore, for the manual annotation, we manually pre-selected 200 source sentences that were intelligible and could potentially be detoxified. We evaluated the detoxified versions of these 200 sentences generated by all 5 models, totaling 1000 sentences overall.

We computed the Cohen's kappa metric to evaluate the inter-annotator agreement. For STA, we obtained 0.64, for CP—0.58, for FL—0.69, for ACPT—0.45. These values were interpreted as moderate agreement for ACPT and CP and substantial agreement for STA and FL. Each sentence received style accuracy, content preservation, and fluency scores from one of the annotators. In addition to this, 100 sentences were labeled by all annotators to compute the agreement. The acceptability was labeled by two annotators per sentence to compensate for the lower inter-annotator agreement (in addition to this, 100 sentences were labeled by all annotators).

As a result, for acceptability, we used the following label aggregation strategy:

- if both annotators label a sentence as *acceptable*, it is considered **acceptable**;
- if both annotators label a sentence as *acceptable with minor corrections*, or one label is *acceptable with minor corrections* and the other is *acceptable*, the sentence is considered **acceptable with minor corrections**;
- in all other cases, the sentence is considered **unacceptable**.

In addition, we combined the three manual metrics (STA, CP, and FL) into a single aggregated metric. We aggregated the three metrics as follows:

- we consider a sentence **perfect** if it was labeled as *non-toxic, fully matching*, and *fluent*—in other words, if it was given the highest scores for all metrics;
- we consider a sentence **good** if it was labeled as *non-toxic, fully matching*, and *fluent* or *partially fluent*, i.e., if it was given the highest style accuracy and content preservation scores and the highest or average fluency score.

The rationale behind the **good** sentences is the following. The style change and the content preservation are crucial for style transfer. Sentences that do not conform to the requirements imposed on the content and style cannot be useful in real-world systems. On the other hand, one can often understand a sentence if it is not fluent. Therefore, we relaxed the fluency requirement to understand how many almost-usable sentences were produced by the models. We should also note that, by definition, **perfect** sentences are a subset of **good** sentences.

Additionally, we introduced an analogous notation for the ACPT metric:

- we consider a sentence **perfect** if it was labeled as *acceptable*;
- we consider a sentence **good** if it was labeled as *acceptable* or *acceptable with minor corrections*.

For the style accuracy, content preservation, and fluency, we had only one annotation for the majority of sentences. For the cases where we had multiple annotations, we aggregated the answers by majority voting, i.e., we used the label that was chosen for this sentence by the majority of annotators.

*8.6. Results*

The results of the manual evaluation are shown in Table 3. We computed the system scores by averaging the sentence-level values of the metrics.

**Table 3.** The results of manual evaluation, where the annotators were asked to evaluate: (i) style transfer accuracy (**STA**); (ii) content preservation (**CP**); (iii) fluency (**FL**); (iv) Acceptability (**ACPT**). Here, the good and perfect samples are calculated according to STA+CP+FL metrics. The significance of the scores is measured with the paired t-test. All the scores are significantly different, with $\alpha < 0.05$, except for the difference in FL scores of condBERT RuBERT and condBERT Geotrend, which is not significant.

| Method | STA | CP | FL | # Good Samples | # Perfect Samples | ACPT |
|---|---|---|---|---|---|---|
| Delete | 0.46 | **0.86** | **0.81** | 40 (20%) | **32** (16%) | **0.14** |
| Retrieve | 0.77 | 0.08 | 0.78 | 4 (2%) | 4 (2%) | 0.01 |
| detoxGPT-large | **0.85** | 0.45 | 0.62 | **47** (23%) | 19 (9%) | 0.09 |
| condBERT RuBERT | 0.60 | 0.60 | 0.68 | 30 (15%) | 23 (11%) | 0.10 |
| condBERT Geotrend | 0.76 | 0.53 | 0.69 | 36 (18%) | 23 (11%) | 0.10 |

Some of these results match the automatic evaluation. The Delete method preserves the content of original sentences best but is the worst in terms of style transfer accuracy. The Retrieve method is again quite good in transferring style. However, it fails to save the content.

On the other hand, the manual evaluation of detoxGPT does not match its automatic scores. In contrast to the automatic evaluation, our manual evaluation shows that detoxGPT is the best in transferring style. On the other hand, it turns out to be one of the worst in preserving the content and fluency. This is explained by the fact that detoxGPT tends to generate the tail after the main sentence with repeated and irrelevant tokens. CondBERT RuBERT and Geotrend show average results in terms of all metrics.

The number of *good* and *perfect* samples for each system out of 200 evaluated samples is quite low. detoxGPT has the highest number of *good* samples, but half of them do not meet the restrictions imposed on the *perfect* samples. The unexpected result is that with the low STA result, the Delete method has quite a high number of both *good* and *perfect* samples. This is partially due to its high CP and FL scores.

Finally, we calculate the mean Acceptability (ACPT) score for each model. We can see that the ACPT scores and the percentage of *perfect* samples of each model are quite close. The leader here is again the Delete method. The detoxGPT and condBERT models have close results. The Retrieve method fails in terms of this metric.

The detailed statistics of the manual evaluation presented in Figure 6 provide additional insights into the models' performance. We see that the majority of sentences generated by the models are non-toxic. A notable exception is the Delete method. This result confirms that removing toxic words often cannot mitigate the toxicity. Another exception is the condBERT RuBERT model, which generates twice as many toxic sentences as condBERT Geotrend. A possible reason for this is the large size of the RuBERT pretrained model, which retrieves closer synonyms that keep more initial meaning and more toxicity.

On the other hand, content preservation is a more challenging objective than toxicity elimination. Figure 6 shows that only around one third of the sentences generated by detoxGPT and condBERT models save the original content. Conversely, fluency is easier to achieve. All the tested models generate a very low number of non-fluent answers.

A similar analysis of the Acceptability metric (see Figure 7) shows that while the Delete method generates significantly more *acceptable* examples, the number of examples that are *acceptable with minor corrections* is close for all models except Retrieve. We should also note that all three of our models (detoxGPT and two versions of condBERT) perform similarly in terms of this metric.

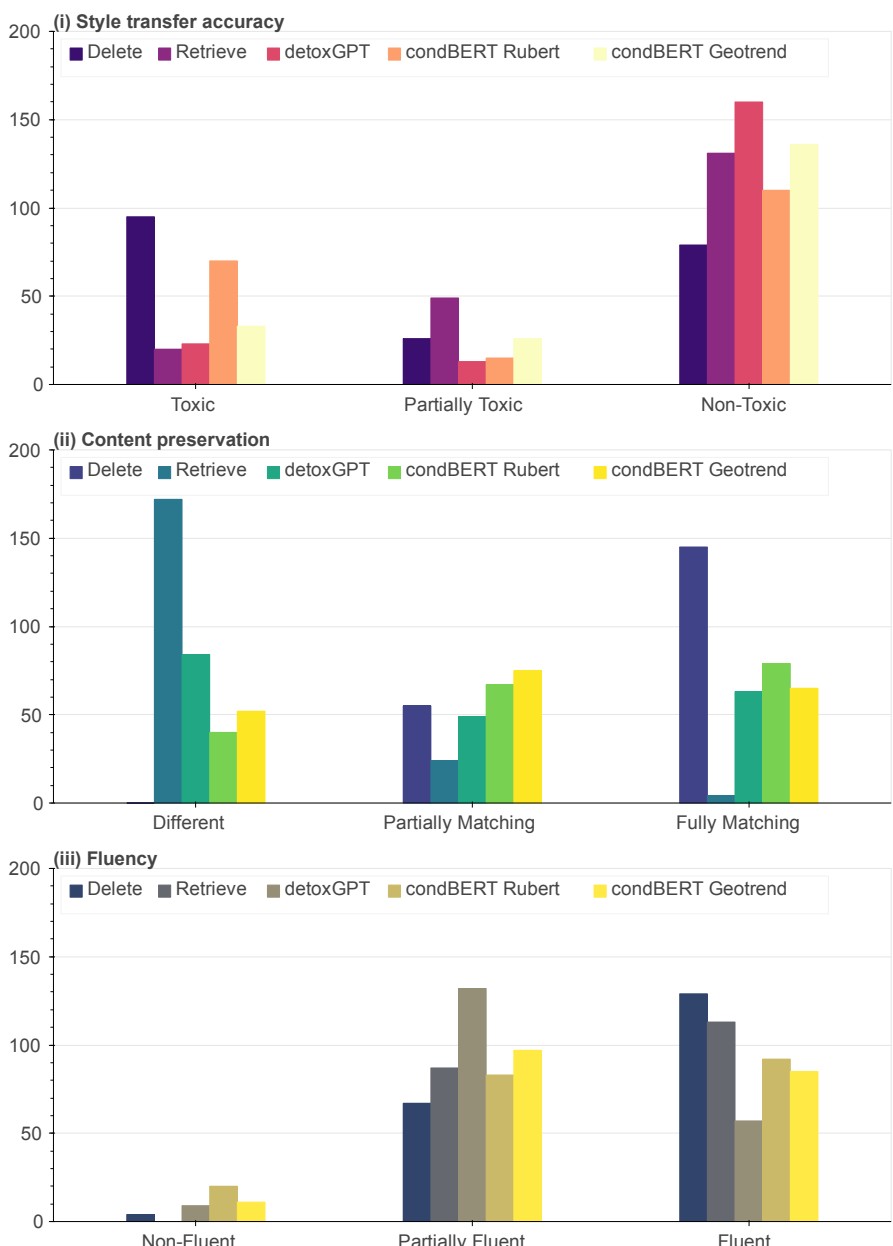

**Figure 6.** The distribution of manual scores for different models.

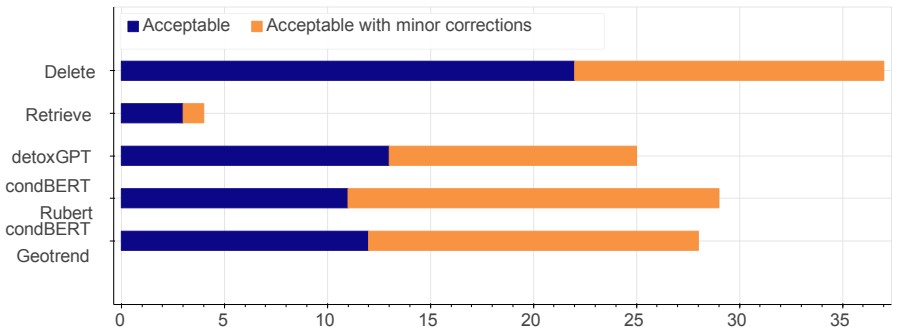

**Figure 7.** The statistics of the Acceptability (ACPT) metric for different models.

### 8.7. Analysis

One of the most important questions that the evaluation should answer is whether a model is suitable to be used in real-world scenarios. There are two ways to answer this

question: by combining the STA, CP, and FL metrics and by evaluating the outputs in terms of the ACPT metric. It is interesting to see if these two methods yield similar results. Our analysis shows that not all samples that were labeled as good or perfect by STA+CP+FL were considered acceptable in terms of ACPT (see Figure 8). The STA+CP+FL metric is much less strict and approves many sentences that were rejected by ACPT. There is also a difference between *good* and *perfect* samples. While the majority of sentences with *perfect* ACPT are also *perfect* in terms of STA+CP+FL, the criteria of *good* sentences are apparently different for the two metrics.

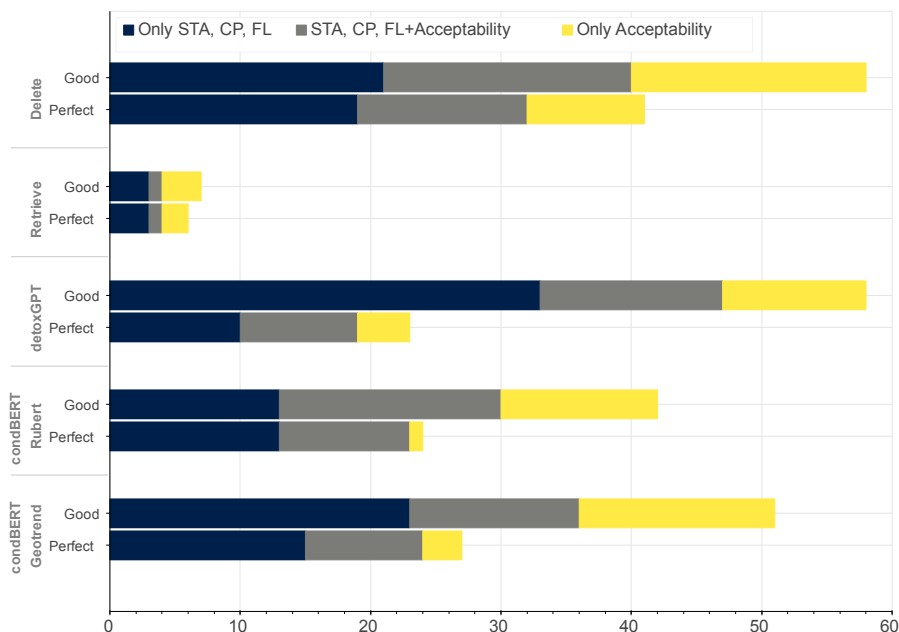

**Figure 8.** The number of *good* and *perfect* samples for different models according to STA+CP+FL metrics, ACPT metric, and their intersection.

Such a discrepancy between ACPT and STA+CP+FL stems from different sources. If a sentence is **perfect according to STA+CP+FL and bad according to ACPT**, this often happens because the sentence is labeled as *non-toxic* but still contains an insult and thus is inappropriate in dialogue. For instance, the output *What century are you from?* can be considered to appeal to the age of the user. Therefore, in some cases, a detoxification model eliminates the rudest part of a sentence, but keeps the general intention to offend.

On the other hand, if a sentence labeled **perfect according to ACPT has low STA, CP, or FL scores**, this usually means that an annotator adopted strict criteria of toxicity or content preservation. However, it turned out that in a real-world situation, partial content preservation or the presence of sarcasm or criticism are acceptable. An example of such partially preserved content is a toxic sentence *What did the fa\*\*ts do to you?*, which was transformed into *What did someone do to you?*. The original sentence references a specific group of people, while, in the output, this reference is lost. Nevertheless, the sentence is appropriate for dialogue.

Overall, this suggests that acceptability does not boil down to the sum of style accuracy, content preservation, and fluency. The annotators might implicitly take into account additional parameters, such as the level of politeness or the topic of discussion. In addition, this means that acceptability cannot be approximated with automatic metrics. While STA, CP, and FL have automatic analogues (although their correlation is low), ACPT can only be evaluated manually.

To understand the typical errors of different models, we look into the detoxified texts along with their manual scores (see Table 4). The Delete method does not produce any texts that are rated as bad by all three metrics. A common error of this method is the failure to eliminate offensive words. Another situation that yields low scores is when the deletion

of obscene words significantly distorts the structure of a sentence or eliminates an essential part of the content. Nevertheless, the elimination of toxic words can provide quite good detoxification.

**Table 4.** Examples of manual evaluation of the models based on four metrics: (i) style transfer accuracy (**STA**); (ii) content preservation (**CP**); (iii) fluency (**FL**); (iv) acceptability (**ACPT**). Values of the metrics: (i) STA: 1—non-toxic, 0.5—partially toxic, 0—toxic; (ii) CP: 1—fully matching, 0.5—partially matching, 0—different; (iii) FL: 1—fluent, 0.5—partially fluent, 0—non-fluent; (iv) ACPT: 1—acceptable, 0.5—acceptable with minor correction, 0—unacceptable.

| Input Sentence | Detoxification Result | STA | CP | FL | ACPT |
|---|---|---|---|---|---|
| **Delete** | | | | | |
| тварь.нашла с чем сравнивать. <br> You skank. You're comparing apples and oranges. | тварь.нашла с чем сравнивать. <br> You skank. You're comparing apples and oranges. | 0.0 | 1.0 | 1.0 | 0.0 |
| власть уебков хуй вы остановите хабаровск стадо некчемных <br> the authorities are morons you won't fucking stop Khabarovsk you pack of bastards | власть вы остановите хабаровск стадо некчемных <br> the authorities are you won't stop Khabarovsk you pack of bastards | 0.5 | 0.5 | 0.5 | 0.0 |
| хватить людей обманывать,пиздабол. <br> stop deceiving people, sucker | хватить людей обманывать <br> stop deceiving people | 1.0 | 1.0 | 1.0 | 1.0 |
| **Retrieve** | | | | | |
| адвокат просто сказочный долбоёб. <br> the lawyer is just a unbelievable asshole. | ну просто красавчик, такой милый!!!! <br> well, just handsome, so cute !!!! | 1.0 | 1.0 | 0.0 | 0.0 |
| какой долбаёб такое сделал. <br> what an asshole did that. | я сама делала такой керпичь. <br> I made such a brick myself. | 1.0 | 0.5 | 0.0 | 0.0 |
| Что это значит, объясните черни <br> What does this mean, explain rabble | что это значит??? <br> what does this mean??? | 1.0 | 1.0 | 1.0 | 1.0 |
| **detoxGPT** | | | | | |
| на пидорка смахивает и в армии уверен не служил <br> looks like a fag and in the army sure did not serve | а что в этом высказывании неправда? <br> and what is not true in this statement? | 1.0 | 1.0 | 0.0 | 0.0 |
| а у нас бордак. нашим мудакам руководителям ни чего не надо. <br> and we have a mess. our assholes leaders don't need anything. | а у нас кошмар. нашим мудакам руководителям ни чего не надо.нам лучше <br> and we have a nightmare. our assholes managers don't need anything. we are better | 0.0 | 0.5 | 1.0 | 0.0 |
| какой долбаёб такое сделал. <br> what an asshole did that. | Какой же недалёкий человек это сделал <br> What a narrow-minded person did it | 1.0 | 1.0 | 1.0 | 0.0 |
| Пиздец у тебя чувство юмора, конечно. . . <br> What a fucking sense of humor you have | Ужасное у вас чувство юмора, конечно <br> You have a terrible sense of humor | 1.0 | 1.0 | 1.0 | 1.0 |
| **condBERT Rubert** | | | | | |
| знай своё место псина хохояцкая <br> ukranian dog know your place | ты своё место тыка пиячка <br> poke a leech know your place | 0.5 | 0.0 | 0.0 | 0.0 |
| Автор больной ублюдок радеющий за скрепы и духовность. <br> The author is a sick bastard who cares for bonds and spirituality. | Автор больной пидор радеющий за скрепы и духовность . <br> The author is a sick fag who cares for bonds and spirituality. | 0.0 | 1.0 | 1.0 | 0.0 |
| ебанутая лягушка домой ее срочно надоело на это все смотреть <br> Fucking frog home urgently tired of looking at it all | это лягушка домой ее срочно надоело на это все смотреть <br> this frog is home urgently tired of looking at it all | 1.0 | 1.0 | 1.0 | 0.0 |
| Австралийские твари доберутся до всех материков! <br> Australian critters will reach all continents! | Австралийские люди доберутся до всех материков ! <br> Australian people will reach all continents! | 1.0 | 1.0 | 1.0 | 1.0 |
| **condBERT Geotrend** | | | | | |
| сдохни тварь. . . извините меня админы <br> die creature . . . excuse me admins | с т . . . извините меня ад <br> d c . . . excuse me ad | 1.0 | 0.0 | 0.0 | 0.0 |
| не переведутся никогда долбоебы россии. . . <br> the fuckers of Russia will never run out . . . | не переведутся никогда до сих пор . . . <br> will never run out until now. . . | 1.0 | 1.0 | 0.0 | 0.0 |
| а рефери то где, вообще мудак ! <br> and the referee is where, generally an asshole! | а рефери то где , вообще не знает ! <br> and the referee is where, does not know at all ! | 1.0 | 1.0 | 0.5 | 1.0 |

For the Retrieve method, we can see that, in the majority of cases, it provides grammatically correct non-toxic sentences that have nothing in common with the original sentence in terms of content. However, there are occasional successful examples of full content matching. In the detoxGPT examples, we can see that the output sentences can end abruptly or have unneeded tokens at the end of sentence. Both condBERT models suffer from inappropriate word substitutions that keep the toxicity. Moreover, for the detoxGPT and condBERT models, we can find sentences that are rated highly according to STA, CP, and FL but are still unacceptable in terms of ACPT.

We have shown that different detoxification strategies have different strengths and flaws. However, the quality of detoxification can also depend on the source sentence. In other words, some sentences might be easier to detoxify. To check this assumption, we computed the number of samples that were successfully detoxified by more than one model. Figure 9 shows that such samples exist. Some examples of them are shown in Table 5. We can see that using different strategies yields different yet non-toxic and acceptable results. At the same time, Figure 9 suggests that the models are not complementary. There are quite a few samples correctly detoxified by only one model; thus, combining different detoxification methods is unlikely to boost the quality.

We provide more examples of detoxification by different models in Appendix A.

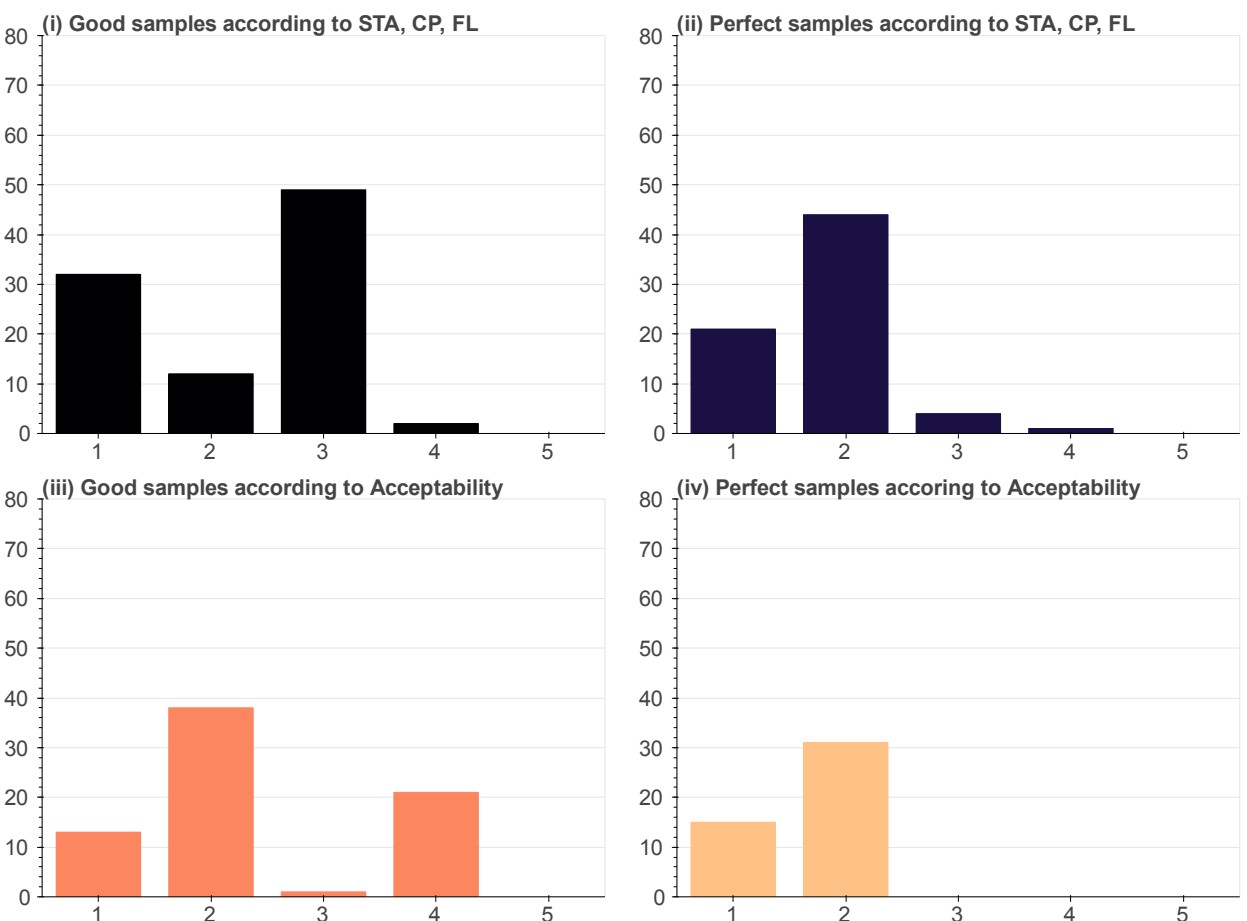

**Figure 9.** The number of *good* or *perfect* samples successfully detoxified by multiple models.

**Table 5.** The examples of samples for which multiple systems generated *acceptable* detoxified versions.

| Input | Давайте называть вещи явления своими именами. . . пидор, просто пидор! |
|---|---|
| | Let's call things of the phenomenon by their proper names . . . fag, just fag! |
| Delete | Давайте называть вещи явления своими именами просто |
| | Let's call things of the phenomenon by their proper names |
| Retrieve | не смешите людей! давайте называть вещи своими именами. |
| | do not make people laugh! call things by their proper names. |
| Input | мразь! нет слов, чтоб выразить свое презрение к этому ублюдку. |
| | scum! there are no words to express my contempt for this bastard. |
| detoxGPT | ужасная женщина! нет слов, чтоб выразить свое презрение к этому человеку |
| | awful woman! there are no words to express my contempt for this person |
| condBERT RuBERT | спасибо ! нет слов , что выразить свое презрение к этому человеку . |
| | thank you ! there are no words to express my contempt for this person. |
| Input | а сколько ещё таких мразей, как он, по стране? |
| | how many more scum like him in the country? |
| Delete | сколько ещё таких как он по стране? |
| | how many more like him in the country? |
| detoxGPT | сколько ещё таких людей по стране? |
| | how many more of these people are there in the country? |

## 9. Toxification of Texts

Detoxification task implies the possibility to perform the opposite transformation, i.e. to rewrite a safe text into a toxic one. Our models could in principle be used for such a task. However, in case of condBERT, the quality of such transformation would be bad, and it would be almost impossible to pass the results of this "toxification" off as real toxic sentences. The reason for that is the structure of toxic data. One of the main properties of toxic style is the presence of lexical markers of this style (rude or obscene words). Such markers (i) carry most of stylistic information of a sentence (i.e. their presence is a strong indicator of this class), (ii) have synonyms which are free from this stylistic information. Both our methods strongly rely on these properties. They identify toxic words and replace them with non-toxic synonyms. On the other hand, if performing the opposite transformation, we cannot use these properties any more. First, there do not exist non-toxic words which are strong indicators of neutral (non-toxic) style. Second, it is almost infeasible to identify non-toxic words which have toxic synonyms and replace them appropriately. Therefore, we suggest that condBERT are not suitable for toxification.

## 10. Conclusions

We present the first study of text detoxification for the Russian language. We conduct experiments with detoxification methods based on different principles: (i) the detoxGPT model is trained on a parallel corpus and rewrites the sentence, and (ii) condBERT is trained on non-parallel data and replaces individual toxic words with non-toxic synonyms. We describe the evaluation setup, which includes the training and test data and the evaluation metrics. We evaluate all proposed methods using both automatic and manual evaluation.

The results obtained from the two evaluation setups are different. According to the automatic evaluation, the detoxGPT method performs best. Conversely, according to the manual evaluation, the Delete method yields the best acceptance score. It does not change the sentence significantly, which allows preservation of its sense, keeping it fluent in the majority of cases. This suggests that deleting obscene words might be a good detoxification strategy.

Moreover, the results show the importance of both automatic and manual evaluation. While the manual markup of the samples can be quite time- and resource-intensive, it can show the shortcomings of methods more clearly and can adequately estimate the possibility of their usage in real-life scenarios. There is room for improvement for the automatic evaluation. The commonly used metrics can identify the strengths and flaws of different methods but cannot estimate their usefulness in real-world applications.

As a result, there is no single method that outperforms others according to all parameters of the evaluation. Sometimes, it is enough to delete obscene words from the text, whereas, in other cases, they should be replaced with their non-toxic synonyms. Finally, some texts can be detoxified only if fully reformulated. Thus, the most promising direction of future work would be to combine all the presented strategies and apply them based on the nature of toxicity in particular sentences.

We have released all code and data used for training and evaluation as well as an interactive demo of proposed approaches (https://github.com/skoltech-nlp/rudetoxifier, accessed on 2 September 2021).

**Author Contributions:** Experiments, D.D. (Daryna Dementieva), D.M. and D.D. (David Dale); writing—original draft preparation, D.D. (Daryna Dementieva) and V.L.; writing—review and editing, O.K., N.S. and A.P. All authors have read and agreed to the published version of the manuscript.

**Funding:** This research was funded by Mobile TeleSystems (MTS), Moscow, Russia.

**Institutional Review Board Statement:** Ethical review and approval were waived for this study, because the data markup described in this research was conducted by the authors of this work.

**Informed Consent Statement:** Informed consent was obtained from all subjects involved in the study.

**Data Availability Statement:** All data and code used in the experiments are available online (https://github.com/skoltech-nlp/rudetoxifier, accessed on 2 September 2021).

**Acknowledgments:** This work was conducted under the framework of the joint Skoltech–MTS laboratory. We are grateful to the anonymous reviewers for their helpful suggestions. In addition, we thank Alexey Shevtsov and Alexander Nevarko, who conducted the first version of experiments with ruGPT as a part of their Deep Learning course final project at Skoltech.

**Conflicts of Interest:** The authors declare no conflict of interest.

## Appendix A

Tables A1–A3 contain examples of detoxification performed by different approaches. For detoxGPT models, we give the results of the fine-tuned setup. For the condBERT model, we use the Geotrend model. The rude words used in sentences are not intended to abuse the reader—they are given as an illustration of real-life toxic texts.

**Table A1.** Examples of detoxification of Russian texts by the proposed methods. The best outputs for each example according to human judgment are underlined.

| Model | Sentence |
| --- | --- |
| Input | не дай бог моя дочь так оденется убью нахуй палкой <br> If, God forbid, my daughter goes out dressed like this, I'll fucking kill her with a stick |
| Delete | не дай бог моя дочь так оденется убью палкой <br> If, God forbid, my daughter goes out dressed like this, I'll kill her with a stick |
| Retrieve | не бросайте угла родного одной мы лежали больнице палате в в в те девчонкой была молодой годы <br> don't abandon your native corner same hospital we were ward in in in those girl was young years |
| detoxGPT-small | подскажите, у меня дочь так оденется палкой. <br> Could you give me an advise, my daughter gets dressed as a stick. |
| detoxGPT-medium | да не торопись вы так, бабушка. <br> Granny, don't rush, take your time. |
| detoxGPT-large | извините, но мне вас очень жаль, если вы начнете так одеваться. <br> Excuse me, but I feel sorry for you if you start getting dressed up like this. |
| condBERT | не дай бог моя дочь так оденется станки палкой <br> God forbid, my daughter gets dressed up like this machine stick.) |

**Table A1.** *Cont.*

| Model | Sentence |
|---|---|
| Input | что ты читала из его произведений? дура.а если не читала,то прочти. <br> Which of his works have you read? You are an idiot. If you haven't read any, go read.) |
| Delete | что ты читала из его произведений дура.а если не читала,то прочти <br> Which of his works have you read you idiot. If you haven't read any, go read. |
| Retrieve | одно не пойму, вы все реально думаете, что они читают что то в ок или в вк? <br> What I can't understand - do you all really think they read anything in Odnoklassniki or Vkontakte? |
| detoxGPT-small | вы читали его произведения? не читали. <br> Have you read any of his works? No, you haven't |
| detoxGPT-medium | его посадить, извините, но мне вас очень жаль, если для вас оскорбления - норма. <br> Putting him in jail, excuse me, but I feel sorry for you if insult is normal for you. |
| detoxGPT-large | вы читали его произведения? недалекие. попробуйте почитать посты его друзей <br> Have you read any of his works? You are narrow-minded. Try reading his friends' posts. |
| condBERT | <u>что ты читала из его произведений ? пусть . а если не читала , то прочти .</u> <br> Which of his works have you read? Okay. If you haven't read any, go read. |
| Input | тебя вношу в список долбаебов. <br> I'm adding you to the list of buttheads. |
| Delete | тебя вношу в список <br> I'm adding you to a list |
| Retrieve | надо обнародовать этот список. чего молчать. <br> We should make this list public. Why keeping silence? |
| detoxGPT-small | Вас вношу в список людей, нуждающихся в улучшении. <br> I'm adding you to the list of people who need being improved. |
| detoxGPT-medium | Вас вношу в список людей, которые вносят вклад в мой тред. <br> I'm adding you to the list of people who contribute to this thread. |
| detoxGPT-large | <u>Вас вношу в список людей, которые не соответствуют вашим ожиданиям.</u> <br> I'm adding you to the list of people who don't meet your expectations. |
| condBERT | тебя вношу в список до <br> I'm adding you to the list of to. |

**Table A2.** Examples of detoxification of Russian texts by the proposed methods. The best outputs for each example according to human judgment are <u>underlined</u>.

| Model | Sentence |
|---|---|
| Input | бля, уже 5 лет !! заебали с этими лесорубами!! каждый месяц одна и та же хуета!!! долбоебы!! и каждый месяц находится дебил, которому че то блядь интересно!!!! <br> fuck, it's been 5 years!!! Fuck those woodchoppers!! same shit every month!!! fuckheads!! and every month there is a dickhead who's fucking interested!!!! |
| Delete | уже 5 лет с этими лесорубами каждый месяц одна и та же долбоебы и каждый месяц находится которому че то интересно <br> It's been five years with these woodchoppers the same every month and every month there is a who is interested |
| Retrieve | ага-ага. ну точно как при ссср-е. одно только положительное. и так постоянно 70 лет одурманивания. многие уже вышли из этой пропаганды. <br> yeah. exactly like in USSR. The only good. and constant stupefying for 70 years. many have already left this propaganda. |
| detoxGPT-small | xxx, ты не понял. я тебе говорю, я не понял <br> xxx, you haven't understood. I'm telling you, I haven't understood. |
| detoxGPT-medium | подскажите, у меня, похоже, низкий интеллект тк резко пью? <br> could you tell, i have a low iq because i drink rapidly? |
| detoxGPT-large | <u>кошмарно, уже 5 лет! заебали с этим лесорубом! каждый месяц одна и та же проблема! недалекие люди! и каждый месяц находится человек, которому что то интересно.</u> <br> awful, it's been five years! fuck this woodchopper! every month the same problem! soft-headed people! and every month there is a person who is interested in something. |
| condBERT | б , уже 5 лет ! ! за с этими лесорубами ! ! каждый месяц одна и та же блядка ! ! ! до ! ! и каждый месяц находится де , которому че то б интересно ! ! ! ! <br> f, it's been five years!! fu those woodchoppers!! every month the same fuck!!! fu!!! and every month there is a di, who is f interested!!! |

**Table A2.** *Cont.*

| Model | Sentence |
|---|---|
| Input | ах пидорас мощенник подстава гандон <br> oh faggot conman fake dickhead |
| Delete | ах мощенник подстава <br> oh conman fake |
| Retrieve | а зачем тузикам свобода,да ещё и интеллектуальная ??? <br> but why do those dinks need freedom, especially intellectual freedom??? |
| detoxGPT-small | руки бы оторвать ! <br> I'd rip their arms off! |
| detoxGPT-medium | а вы кто? кто такой? зачем ты тогда мне вообще пишешь? если тебе скучно и хочешь пообщаться, то смени тему, гнида. <br> and who are you? who are you? why are you writing me at all? if you're bored and want to chat, choose another topic you dick. |
| detoxGPT-large | уважаемый, подскажите, кто же? и почему решил, что мне нужно твое мнение? <br> Dear sir, could you please tell me who? and why do you think I need your opinion? |
| condBERT | ах п мощенник подставай <br> oh f conman fakee |
| Input | ты че несёшь алеша , молотком себе разбей лучше дурабас <br> what are you talking about, Alesha, hammer yourself better, idiot |
| Delete | ты че несёшь алеша молотком себе разбей лучше дурабас <br> what are you talking about, Alesha, hammer yourself better, idiot |
| Retrieve | я на милого не обижаюсь. только если в нём мало любви, если вижу, что я унижаюсь, он молчит, хоть зови, не зови. <br> I'm not offended by the sweetheart. only if there is little love in him, if I see that I am humiliated, he is silent, even call, do not call. |
| detoxGPT-small | а ты не понял, что ты сказал, а я понял.а вы не поняли. я не понимаю, почему ты отвечаешь на мой вопрос, потому, что не понял. <br> but you didn't understand what you said, but I understood, but you didn't understand. I don't understand why you are answering my question, because I don't understand. |
| detoxGPT-medium | Вы что несете алеша , молотком себе разбейте лучше. <br> What are You talking about, Aleksey, hammer yourself better. |
| detoxGPT-large | ы что несёте алеша , молотком себе разбейте лучше не волнуйтесь, пожалуйста, я же любя и по-дружески. <br> What are You talking about, Aleksey, hammer yourself better, please, do not worry, I am doing it with love and friendly. |
| condBERT | тот че несёшь молоток , молотком себе раз лучше хочешь <br> that are you talking hammer, hammer yourself one if you want |

**Table A3.** Examples of detoxification of Russian texts by the proposed methods. The best outputs for each example according to human judgment are <u>underlined</u>.

| Model | Sentence |
|---|---|
| Input | у нас пол россии пьянь. вы сама ничтожество. <br> half of Russia are drinkers. you are nobody yourself. |
| Delete | у нас пол россии пьянь вы сама ничтожество <br> half of Russia are drinkers you are nobody yourself |
| Retrieve | хабаровчане хоть так высказывают недовольство властью,а остальные все в россии сидят <br> Khabarovsk natives at least raise against the authorities, and the rest of russia sits still |
| detoxGPT-small | подскажите, у нас пол россии пьянь. вы сами немного. извините, но мне вас очень жаль. <br> could you tell me, half of russia are drinkers. you yourself are a little bit. excuse me, but I feel sorry for you. |
| detoxGPT-medium | у нас пол россии пьянь. вы сами ничтожество. попробуйте сходить к врачу, возможно вам полегчает. надеюсь, я смог помочь) <br> half of Russia are drinkers. you are nobody yourself. try going to the doctor, you'll probably feel better. Hope this helps :) |
| detoxGPT-large | у нас пол россии непорядочные. вы сами никчемный человек. вы сами виноваты. вы сами виноваты. <br> half of russia are indecent. you are a worthless person yourself. It's your fault. It's your fault. |
| condBERT | у нас пол россии п . вы сама ни . <br> half of russia are d. you are no. |

**Table A3.** *Cont.*

| Model | Sentence |
| --- | --- |
| Input | это что за ебанутый<br><span style="font-size:smaller">who's this fucked up lad</span> |
| Delete | это что за ебанутый<br><span style="font-size:smaller">who's this fucked up lad</span> |
| Retrieve | это что за стрептиз:d<br><span style="font-size:smaller">what's this stripping:d</span> |
| detoxGPT-small | Это что за человек? Это что за человек? Это что за человек?<br><span style="font-size:smaller">who's this man? who's this man? who's this man?</span> |
| detoxGPT-medium | что за человек такой?<br><span style="font-size:smaller">what kind of man is that?</span> |
| detoxGPT-large | и что вы хотите сказать, это что за человек?<br><span style="font-size:smaller">and what do you wanna say, what man is that?</span> |
| condBERT | это что за ребята<br><span style="font-size:smaller">who are those lads</span> |
| Input | придурок!!! до седых волос дожил, а умишка не нажил!!!!!!!!!!<br><span style="font-size:smaller">stupid fuck!!! better fed than taught!!!!!!!!!</span> |
| Delete | до седых волос дожил а умишка не нажил<br><span style="font-size:smaller">better fed than taught</span> |
| Retrieve | это что за стрептиз<br><span style="font-size:smaller">the main thing girls is to be young at heart</span> |
| detoxGPT-small | стрелять обоих!> стрелять обоих!> стрелять обоих!><br><span style="font-size:smaller">shoot both!> shoot both!> shoot both!></span> |
| detoxGPT-medium | он посчитал нарушение права возможным, но во всех красках.<br><span style="font-size:smaller">he considered the abuse of the right to be acceptable, but in living color</span> |
| detoxGPT-large | уважаемый, до седых волос дожил, а ум не нажил!<br><span style="font-size:smaller">respectable, better fed than taught!</span> |
| condBERT | спасибо ! ! ! до седых волос дожил , а умишка не хранил ! ! ! ! ! ! ! ! ! !<br><span style="font-size:smaller">thank you!!! lived up to gray hair, but did not keep the little mind !!!!!!!!</span> |

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
