# Peer review of "Methods for Detoxification of Texts for the Russian Language†"

_mti, doi:10.3390/mti5090054_

Round 1

Reviewer 1 Report

The paper evaluates two approaches to replacing toxic sentences with non-toxic ones in the Russian language: an approach based on single word replacement using a BERT-based model, and a supervised language generation approach based on a GPT3 model. Both models are automatically evaluated on a single corpus using accuracy, content preservation, and language fluency metrics.

It is great to see a work that attempts to fill the gap in confronting toxic online content in languages other than English. The work compares the performance of two contemporary techniques for style transfer on a Russian corpus. The paper is well written and structured.

My main concerns are the lack of the clear task formulation and the lack of manual evaluation. Even though a formal problem statement is provided in Sec. 2, it does not clearly define the task. What is considered toxic in this work? The field of online toxicity (abuse, offense, hate speech, etc.) has been suffering from the lack of clear and compatible definitions. Every work uses its own definitions, and the datasets are often incompatible. So, at least the paper needs to clearly state what is considered toxic and non-toxic. For the style transfer task: is it enough just to remove profanities? Do we want to preserve the content or the meaning of an utterance? If the meaning of an utterance is just to offend the other person (as in examples in Table 4), is it even possible to replace it with a non-toxic alternative or is it better to remove the whole sentence altogether? Also, it might be helpful to provide examples of sentence re-writings. Further, how can the task be applied in the real world? The paper states that the approach can “offer a user a non-offensive version of this text”. How useful is it? If the user’s intention is to offend the other person, would they use such a service? It may be more useful for unintentional, implicit offenses where users are unaware of the offensive meaning of their texts. But in such cases, different approaches might be needed. Finally, how ethical is it to change the user’s text and the user’s intentions, and potentially introduce further biases (e.g., political, moral, gender-based, etc.)?

The two methods are evaluated using a suite of automatic metrics. While they provide some information on methods’ strengths and weaknesses, they are very limited. The accuracy metric is based on an automatic toxic/non-toxic classifier, which is imperfect. The classifier is trained on the same data on which the classifier for word toxicity weights is trained, which introduces a bias. The content preservation metrics are not very useful if we care about meaning preservation.  The perplexity metric is not very useful either. Further, it is not obvious (and is not motivated in the paper) why the geometric mean of the three metrics is a good combined measure. For example, I would argue that if a sentence remains toxic after re-writing, the task is incomplete even if the re-writing is very fluent. The task requires manual evaluation. Human annotators can be asked if the re-writing is acceptable (in the real-world scenario) to assess the potential usefulness of the current technology. Also, the annotators can compare different re-writings overall and separately by various aspects (fluency, meaning preservation, etc.). I would also argue that more context (sentences before the target one) might be required to make such decisions.

I like one of the conclusions of the study that the choice of the style transfer method may depend on a sentence: for some sentences it is enough to remove or replace an offensive word, and for some the complete re-writing is needed. But we don’t know if and when that is true, and how often each of the situations tend to occur in the real-world communications. It would be beneficial to manually estimate the frequencies of these situations and the potential success of each of the evaluated methodologies.

The paper claims in several places that the presented approaches “solve” the task. Based on the evaluation results and example sentences presented in Tables 2-4, I would say that the task is far away from being solved. None of the presented examples (and I assume they are already cherry-picked) have a good automatically generated non-toxic alternative. So, I strongly recommend to tone down statements like “they can successfully solve the task” (lines 32-33) or “in most cases, at least one of the detoxGPT models provides a sensible answer” (lines 310-311).

Minor comments:

  • Line 8 “unsupervised approach based on BERT”: this approach also requires (non-parallel) labeled data, so I wouldn’t call it ‘unsupervised’.
  • Line 17 “smarter ways”: smarter than what?
  • Line 28 “the work [8] gives …” -> “the survey by Jin et al. [8] provides more examples …”
  • Line 25-27: the sentence needs re-writing
  • Line 71: “ model [15].”
  • Line 154: the word ‘model’ is repeated twice
  • What corpus of non-toxic sentences is used for the Retrieve baseline method?

Author Response

Dear Reviewer,

Thank you for your thoughtful comments and discussions. We believe that we have addressed all your concerns. The details of our work are listed below.

To add more clarifications about the goals of our proposed methodology, we add examples of ideal detoxification in the Introduction section. In addition,  we added the section “Application and Ethics” to address the questions about whether the methods can be useful in real-life scenarios and which ethical issues can appear.

There were concerns about the automated evaluation used in the work. It is true that the toxicity classifier used for style transfer accuracy estimation is imperfect but this is the de-facto current standard for the evaluation of the style transfer approaches which we adopt in our work.  

However, again, the main purpose and the contribution of this work is the first application of the detoxification / style transfer technology to the Russian language. So therefore we deliberately decided to follow the evaluation setup --- however imperfect -- as adopted by the international community.

To address the issue with evaluation,  we added a “Manual Evaluation” section with broad analysis of the results, examples and comparison with automatic evaluation results.

In addition, we toned down the discussion suggesting that these are the first yet preliminary results on the detoxification of the Russian texts.

Finally, we did all the corrections in the text according to your minor comments.

Reviewer 2 Report

The paper is well written and well organised. However, there are a few points that need to be considered in order to improve the quality of the paper:

  • As the paper is written in English, Figure 4 which is the illustration of the main idea of condBERT approach should be modified and translation of the Russian sentences should be added to the figure to make it more understandable.
  • In section 6.1. Datasets, it has been explained that two corpora of toxic comments which are available on Kaggle repository, have been concatenated and a joint corpus called RuToxic dataset has been created. This needs more clarification as these two Kaggle datasets have different structures and formats. Your concatenation approach should be explained clearly and in more details.
  • I would recommend authors to add accuracy percentage and confusion or correlation matrix in order to discuss the general performance of the models. In this way, the readers will be able to compare the performance of the implemented models for Russian language with the existing models for English language.
  • Percentage for non-toxic should be added to line 227 in the paper.
  • In section 6.1. Dataset, it has been explained that a fraction of your dataset has been used to construct the parallel training data for This needs more clarification. Why you have done this? Why you have selected 200 toxic sentences and you manually rewrite them into non-toxic ones. The reason has not been explained clearly and more clarification is required.
  • I would strongly recommend the authors to add some information to the 3. Results and Discussion section about comparison between the implemented models for Russian language and the existing models for English language. This will be very beneficial for the readers and can help them to understand more about reliability of the implanted models.
  • Table 3 and table 4 are in the middle of reference section. This should be modified and the tables should be located in the right place in the body of the paper.

Author Response

Dear Reviewer,

Thank you for your thoughtful comments and corrections. There is a list of actions that were done to address your concerns:

  • We changed the illustration of condBERT approach to English notations;
  • We added more details about RuToxic dataset and its creation;
  • Unfortunately, we did not get the idea about the confusion matrix of the model, as we are not solving the toxicity classification task, but the style transfer task. However, we added a new section dedicated to manual evaluation of the results where we did comprehensive analysis and presented examples of style transfer results and errors.
  • We inserted additional details that were required for the clarification.
  • The comparison with English language requires new research, datasets collection, detoxification models’ tuning and  additional classifier training and metrics implementation. We believe that this is out of the scope of our research. We want to focus on specifically Russian language and more thorough evaluation of the proposed approaches.

Round 2

Reviewer 1 Report

I would like to thank the authors for thoroughly addressing my comments. I especially appreciate the time and effort that went into the manual evaluation.

Minor issues:

  • Table 1, example 1: ‘its’ -> ‘it’s’, ‘not competent’ -> ‘incompetent’

  • Fig. 1b: ‘The prevention of the over-fitted on open data chatbot to be rude to the user and damage the company’s reputation.’ -> ‘Preventing chatbots from being rude to users when trained on open (toxic) data.’

  • Line 91: ‘states’ -> ‘stands’

  • Sec. 2 header: maybe ‘Motivation’?

  • Sec. 3.1: please avoid using references (e.g., [13]) as nouns (e.g., ‘[13] calls …’)

  • Lines 113-117: I disagree. The NLP community is aware of the importance of well-defined categories for data annotation. Many toxic/abusive/hate speech datasets come with carefully crafted annotation guidelines that contain detailed category definitions and examples. Nonetheless, data annotation often remains highly subjective.

  • Definition 3.1: theta function is introduced, but not used. The second condition cannot be satisfied: the content will be changed by the style transfer system.

  • Sec. 8.1: This definition of toxicity can be mentioned in Sec. 3.1 as the one adopted in this work (even though the training data used in this work might have been annotated using slightly different rules). However, this definition is inconsistent. If a sentence has rude words it can be labeled as ’toxic’ and as ‘partially toxic’. It is unclear what is meant by ‘openly toxic’. It seems to be related to explicit abuse, but needs to be clarified. Also, I’m not sure about the intent behind introducing the ‘partially toxic’ category.

  • Sec. 8.2 and 8.3: the questions for annotators are switched.

  • Lines 559: ‘labeled for all annotators’ -> ‘labeled by all annotators’

  • Table 3: are the reported numbers of ‘good’ and ‘perfect’ samples based on the aggregation of STA, CP, and FL metrics or on the ACPT scores?

  • Sec. 8.4: according to the annotation guidelines, a sentence is labeled ‘acceptable’ if the three requirements are met: the sentence is non-toxic, grammatically correct, and matches the original sentence as much as possible. Then, the definitions of ‘perfect’ and ‘good’ sentences should be the same for acceptability and STA+CP+FL, but they are different in Fig. 8. Please, explain.

  • Lines 642, 664: the table numbers (and the tables themselves) are missing. 

Author Response

Dear Reviewer,

Thank you for your comments. We believe we have addressed all your and took into account all your propositions for improvement.

We thoroughly went through the text and did grammar and typo checks. We fixed the definitions of Toxicity and Text Style Transfer to address mentioned issues. We added more clarifications in the Manual Evaluation section to explain the difference between different metrics accumulative results.

Thank you for your time and such a deep dive into our work.